# Burnout Syndrome and Related Factors in Mexican Police Workforces

**DOI:** 10.3390/ijerph19095537

**Published:** 2022-05-03

**Authors:** Irene N. Torres-Vences, Eduardo Pérez-Campos Mayoral, Miguel Mayoral, Eduardo Lorenzo Pérez-Campos, Margarito Martínez-Cruz, Iban Torres-Bravo, Juan Alpuche

**Affiliations:** 1Doctorado en Ciencias en Desarrollo Regional y Tecnológico, Tecnológico Nacional de México, IT Oaxaca, Oaxaca 68030, Mexico; vences_ir@hotmail.com; 2Centro de Investigación Facultad de Medicina UNAM-UABJO, Facultad de Medicina y Cirugía, Universidad Autónoma Benito Juárez de Oaxaca, Oaxaca 68020, Mexico; mianmayo@hotmail.com (M.M.); or pcampos@itoaxaca.edu.mx (E.L.P.-C.); 3Tecnológico Nacional de México, IT Oaxaca, Oaxaca 68030, Mexico; martinezcu9@hotmail.com; 4Red Nacional de Asociaciones Policiales, A.C. Puebla 72480, Mexico; torres_iban@hotmail.com

**Keywords:** burnout, occupational health, police

## Abstract

Burnout (BO) is a response to prolonged exposure to work-related stressors characterized by emotional exhaustion (EE), depersonalization (DP), and reduced personal accomplishment (PA). The police working environment includes continued critical life-threatening situations, violence, and injuries, among other related factors putting them at high risk of distress. The objective of this study was to evaluate the association between Burnout Syndrome and sociodemographic, occupational, and health factors in Mexican police officers. We applied the Maslach Burnout Inventory Human Services Survey (MBI-HSS) to 351 active members of the Mexican police workforce. In addition, a specific questionnaire identified the presence of chronic degenerative diseases, hypertension, diabetes, digestive diseases, self-perception of food quality, and hours of sleep. Furthermore, 23.36% of police workforces presented high levels of burnout; 44.16% of police were highly emotionally exhausted, 49.29% had lost empathy with people, and 41.03% presented low personal achievement. Moreover, the worst levels of the syndrome were present in people with a poor self-perceived health status, poor perception of diet quality, without regular mealtimes, bad sleep habits, and elevated Body Mass Index. Data suggest that in Mexican police officers, BO is dimensionally different from all other groups previously studied (DP > EE > PA).

## 1. Introduction

Leka, Cox, and Griffiths (2004) define work stress as a pattern of psychological, emotional, cognitive, physiological, and behavioral reactions to extremely overwhelming or demanding aspects of the responsibilities, organization, and work environment [1]. Zubiaga (2009) establishes that excessive demands on the individual’s ability to control, work overload, ambiguity and role conflict, overqualification of skills, personal work relationships, lack of recognition, little or no participation in decisions, and job stability uncertainty are critical factors that contribute to work stress [2].

When an individual experiences a prolonged period of stress without the possibility of recovering balance, it could generate somatic and behavioral discomforts. Burnout Syndrome (BO) appears as a specific response to prolonged exposure to work-related stressors [3]. BO is a three-component syndrome: emotional exhaustion (mental or physical fatigue); depersonalization (detachment from work and lack of empathy with people); and reduced personal fulfillment (feelings of inadequacy and insecurity) [4,5]. Burnout can affect both the individual and the organization. In individuals, it can cause emotional problems, anxiety, feelings of helplessness, irritability, feelings of alienation, apathy, aggressive behavior, alcohol dependence, family problems, and physiological alterations, such as cardiovascular, immunological, sexual, and muscular or digestive problems. Regarding the organization, the consequences of BO are related to absenteeism, increases in turnover, and reduction in productivity [6,7].

There are only a few studies about the medical consequences of Burnout Syndrome. However, in a recent work, Bayes et al. (2021) summarize changes in the nervous system, immune system, metabolic, and others [8]. It is expected that all these disorders are significant consequences of police work.

Law enforcement services are one of the largest workforces globally. By 2019, in the European Union, there were 1.49 million police officers [9]; meanwhile, in the United States of America, the were one million police officers [10]. Policing is a challenging full-time occupation; duties include the pursuit of presumed criminals, giving protection to the population, aiding people in distress, and maintaining the safety of streets and communities [11].

The nature of police duties is related to several organizational, administrative, and physical stressors; excessive workloads; inadequate support from supervisors and colleagues; low wages; and constant exposure to traumatic incidents (violence, force use, life-threatening situations) [12,13].

Working conditions affect the mental health of police officers and their interaction with citizens and their families. In the last decade, a significant prevalence of mental health disorders has been found in police officers, such as anxiety, depression, harmful use of alcohol, suicidal thoughts, and behaviors [14,15,16,17,18,19]. In 1975, Schwartz and Schwartz pointed out that police officers constitute one of the professional groups with the highest frequency of suicides [20]. Family is also affected; there is an increased incidence of family problems and higher divorce rates compared to other professional groups studied [21,22].

Physical health has a close relation to working conditions; police officers have a high incidence of hypertension, dyslipidemia, glucose intolerance, ischemic heart disease, sleep disorders, cardiovascular diseases, diabetes, overweight, and obesity compared to the general population [23,24,25,26,27].

In Mexico, police officer’s health conditions, according to the National Survey of standards and police professional training of the National Institute of Statistics and Geography in 2017, were documented as the following: 79.9% were overweight or obese, 18.6% had a chronic disease, 9.3% had hypertension, 6.6% had diabetes, 3% had chronic stress, 1.5% had heart diseases, 1.4% had pulmonary diseases, and other diseases were anemia, hepatic diseases, HIV, cancer, and leukemia. In addition, 53% suffered from a health affectation due to labor-related activities, such as weight gain or loss, irritability, stomachal problems, loss of appetite, stress, cardiovascular problems, loss of control, and depression. Among working conditions, the average weekly working hours was 70, while 70.6% did not have regular mealtimes during working hours, and 59.4% did not have psychological health care; moreover, only 68.6% earned enough income to satisfy their basic needs at home (enough food, medic care, clothing, scholar needs, home maintenance) [28].

Since policing implies a close relationship with citizens, adding to their excessive workload is the high expectations of society regarding the performance of their tasks and precarious and deficient conditions in which they must carry out their work; various studies place the police among the occupational groups particularly exposed to work stress and consequently to BO [29,30]. Burnout Syndrome has been studied in several police forces worldwide; in Spain, 32.2% experienced high levels of BO [31]; in Bulgaria, BO prevalence in correctional officers was 74–53% [32], in Chile, a prevalence of 28% for high BO was reported [33], in North America, the prevalence of BO was 17.7%. Meanwhile, in Mexico, the prevalence was 54.9% among traffic police and 31.1% among preventive police officers in Mexico City.

Mexican police forces are under constant stress, suffer the loss of mental and physical health, and have very difficult working conditions. Therefore, the objective of this study was to assess the prevalence of Burnout Syndrome and evaluate the association between BO dimensions and sociodemographic, occupational, and health factors in the Mexican police. Our work aims to contribute to understanding the development of this syndrome and the related health consequences for police.

## 2. Materials and Methods

To calculate the sample size to determine BO frequency in Mexican police officers, we used the OpenEpi calculator [34]. We used the last data on the police workforces size population of Mexico, the 2015 National Census of Government, Public Security, and Estate Penitentiary System, published by Instituto Nacional de Estadistica y Geografia, which reported 213,048 individuals working in public security tasks (preventive, traffic, auxiliary, administrative, municipal, estate, and directives) [35]. The expected frequency of BO was adjusted to 31.1% since Sanchez-Nieto reported the frequency of having one of three components of the syndrome in a sample of Mexican police officers [36]; the estimated sample size was 329 individuals (confidence limit was 5%, and confidence interval was adjusted to 95%).

Sampling was non-probability by convenience, using an online survey through the SurveyMonkey™ website (survey instrument included as Appendix A); we distributed the survey among State Delegations of the National Network of Police Associations from 30 May 2017, to 10 May 2018.

We applied the Maslach Burnout Inventory Human Services Survey (MBI-HSS), widely used in the public security forces context [6,36,37]. Through twenty-two items, it measures BO’s scale values by adding the items for each dimension. According to the cutoff values proposed by Maslach and Jackson (1997) [38], the level of risk for each dimension was normal, trait, and risk. For emotional exhaustion (EE) and depersonalization (DP), high scores correspond to a high level of burnout. A lower score for personal achievement (PA) leads to increased burnout. The maximum score was 54 points for EE, 30 points for DP, and 48 for RP (Table 1).

Similar studies in Latin America verified the scale’s validity [6,37]. For example, in Mexico, Sanchez-Nieto et al. (2011) used it to evaluate a population of traffic officers in Mexico [36].

To characterize BO in the population, we calculated the mean and standard deviation for each dimension; as proposed by Maslach et al. (1997) [38], individuals were considered BO positive if three dimensions had risk scores (EE > 26, DP > 9, or PA < 34).

### Analysis of Sociodemographic, Work, and Health Factors

We studied sociodemographic (age, marital status, education) and employment factors (rank, work shifts, jurisdiction, functions, seniority) related to stress in the police. In addition, we evaluated the health status (perception of health; the presence of hypertension, diabetes, and digestive diseases; perception of food quality; mealtimes; weight; height; and hours of sleep) characteristics of the population and used them to determine possible associated factors between BO dimensions. Finally, we calculated the Body Mass Index (BMI) using weight and height (kg/m^2^).

We used the Statistical Package for Social Sciences software (OSx, V24, Armonk, NY, USA) for statistical analysis. For descriptive statistics, we calculated central tendencies and variability. To examine possible correlations, we conducted a Pearson’s correlation test. For mean differences between groups, we used Student’s t-test or one-way analysis of variance (ANOVA) using Tukey’s post-test. To elucidate the role of sociodemographic, employment, and health factors on the Burnout Syndrome, we completed an Artificial Neuronal Network Analysis (ANN). We used a multilayer perceptron as a dependent variable and used BO categorized as risk (if all dimensions received high scores) and normal (if not); as factors, we used self-perception of health; the presence of hypertension, diabetes, and digestive diseases; perception of food quality; mealtimes; and hours of sleep; and as covariables, we used age, BMI and seniority. Data partition was established randomly according to relative case numbers of 70% for training and 30% for testing. The architecture was customized by two hidden layers, the hyperbolic tangent activation function, and on the output layer, the activation function was identity. The training type was batch, and the algorithm was optimized using the scale conjugate gradient.

This study was according to the Declaration of Helsinki of the World Medical Association, also respecting the correspondent legislation, using a protocol approved by the Research Committee of the postgraduate program of the Instituto Tecnologico de Oaxaca (000378).

## 3. Results

### 3.1. Mexico Police Workforce Characterization

In this research, we included a total of 351 police officers. The mean age was 39.93 ± 8.66 years old. Most of the participants were married (73.39% CI 67.85–78.93%), 60.48% had college studies (CI 54.36–66.61%), and only 20.56% (CI 15.5–25.63%) had a bachelor’s degree.

In the Mexican police, the average time of service in force was 11.64 ± 8.4 years. The most frequent work shifts were 24 h of work for 24 h of rest (55.82% CI 55.82–62.03%), followed by 12 h/24 h (14.86% CI 10.41–19.32%), 12 h/12 h (10.44% CI 6.62–14.27%), and 8 h/16 h (10.04% CI 6.28–13.0%); mixed schedules are also present in 4.82% of the population, while 4.01% (CI 1.56–6.47%) of the officers reported being confined (24 h active). Regarding the scope of their jurisdiction, 54.84% (CI 48.60–61.08%) of the police officers belonged to the municipal sphere, 36.29% (CI 30.26–42.32%) belonged to the state level, and 8.47% (CI 4.98–11.96%) belonged to the federal level. According to the duties performed in the corporation (functions), 85.89% (CI 81.52–90.25%) of the individuals were operative (fieldwork), and 14.11% (CI 9.75–18.48%) were administrative. Regarding rank, 86.7% of the participating elements belong to the basic levels (police officer, former police, second police, or third police).

To study health factors in the police, we measured the self-perception of health status. A total of 43.17% (CI 35.99–48.34%) of the police officers perceived their health status as very bad to fair; 19.68% (CI 14.71–24.65%) of the population reported suffering from some chronic disease, of which digestive disorders were the most common (30.92%, CI 25.14–36.7%), 17.27% (CI 12.54–22%) suffered from hypertension, and 9.24% (CI 5.62–12.86%) suffered from diabetes. Regarding their nutrition status, 68.95% (62.77–75.13%) of the police officers considered their diet quality very bad to fair. Most of the population (84.36%, CI 79.79–88.88%) did not have regular mealtimes. According to the IMC, 82.26% (CI 77.47–87.05%) were overweight or obese. Regarding the hours of sleep, 83.13% (CI 78.45–87.83%) of the population slept 6 h or less a day (Table 2).

### 3.2. Characteristics of Burnout Syndrome in the Population

The MBI-HSS has a scale for individuals calculated by the mean value of each dimension. The average scale values were 23.29 ± 0.89 for EE, 11.38 ± 0.58 for DP, and 34.63 ± 0.55 for PA (Table 3).

Our data show that 23.36% (IC 95% 18.91–27.81) of the police forces in Mexico were positive for Burnout Syndrome. When we examined scores on each dimension, we found risk scores of EE in 44.16% (IC95% 38.94–49.38) of the population, risk scores of DP in 49.29% (IC 95% 44.03–54.54), and risk scores of PA in 41.03% (IC 95% 35.85–46.20) of the population (Table 3).

We found significant correlations in three dimensions: emotional exhaustion was correlated with seniority (r = 0.252, *p* > 0.001) and BMI (r = 0.219, *p* > 0.001), depersonalization correlated with seniority (r = 0.135, *p* > 0.05) and BMI (r = 0.152, *p* > 0.001), and personal achievement correlated with seniority (r = −0.160, *p* > 0.05) and BMI (r = −0.228, *p* > 0.001). For self-perception of health status and diet quality, we found direct correlations to personal achievement (r = 0.387, *p* > 0.001), (r = 3.51, *p* > 0.001); meanwhile, inverse correlations were found for emotional exhaustion (r = −0.569, r = −0.441, *p* > 0.001) and depersonalization (r = −0.555, r = −0.0428. *p* > 0.001) (Table 4).

When comparing BO dimensions with marital status, we found that depersonalization presented higher values (16.9 ± 10.20) for divorced personnel than those who were married. Work shifts, jurisdiction, functions, and rank were not related to BO dimensions.

We explored the mean differences of BO dimensions on chronic diseases and found higher values for EE (31.20 ± 15.26 vs. 22.74 ± 17.38) and DP (16.22 ± 8.94 vs. 11.02 ± 9.89) and lower values for PA (32.29 ± 9.24 vs. 35.64 ± 10.13). For the presence of hypertension, we found higher values for EE (30.44 ± 75.25 vs. 23.15 ± 17.07) and DP (15.95 ± 10.28 vs. 11.23 ± 9.66) and lower values for PA (32.21 ± 10.75 vs. 35.56 ± 9.80). The presence of diabetes was only related to higher values of EE (31.78 ± 15.15 vs. 23.65 ± 17.35). For the existence of digestive disorders, we found higher values for EE (33.68 ± 15.88 vs. 20.23 ± 16.34) and DP (17.43 ± 9.52 vs. 9.66 ± 9.15) and lower values for PA (31.23 ± 10.33 vs. 36.70 ± 9.46) (Table 5).

When comparing BO to regular mealtimes, we found better values in EE (18.67 ± 16.64 vs. 25.47 ± 17.23), DP (8.51 ± 8.86 vs. 12.70 ± 9.98), and PA (39.13 ± 9.59 vs. 34.21 ± 9.94). When we compared sleep quality by hours of sleep per day, there was a statistical difference between groups, as determined by one-way ANOVA for EE (F = 9.99, *p* < 0.001), DP (F = 8.57, *p* < 0.001), and PA (F = 10, *p* < 0.001). A Tukey post hoc test revealed that all dimensions were significantly lower when sleeping hours were 6–9; for EE, 14.83 ± 16.57 (*p* < 0.001); DP, 7.29 ± 9.16, (*p* < 0.01); and PA, 40.90 ± 7.77 (*p* < 0.001). No statistical differences was found between <4 h and 4–6 h, for EE (*p* = 0.160), for DP (*p* = 0.076), and for PA (*p* = 0.551) (Table 5).

### 3.3. Multilayer Perceptron Neuronal Network of BO and Related Factors

To analyze the relation between sociodemographic, working, and health factors and BO syndrome, we used MLP-ANN. The input factors were the self-perception of health; the presence of hypertension, diabetes, and digestive diseases; the perception of food quality; mealtimes; hours of sleep; age; BMI; and seniority, yielding a total of 26 units. After the first hidden layer, the number of units was 5, and after the second hidden layer, the number of units was 4. The output layer was BO, with two units (computed model included as Appendix A). In training, we obtained a sum of squares error (SSE) of 24.6, and in the test, the SSE was 7.403, with 15.20% being incorrect predictions. Overall, 84.8% were correctly classified, with over 81.30% during training.

From the model obtained, the most important independent variables were BMI (0.295); the higher values were related to BO, and lower age contributed to BO (0.223). Meanwhile, the worst health status contributed to being the third most important variable (0.127), and less critical were fewer sleeping hours (0.02) and the presence of chronic diseases (0.015) (Figure 1).

## 4. Discussion

Mexican police workforces are mainly men of the national average age (39 y.o.), married, college-educated, and with 11.64 years of seniority. Furthermore, they work over 84 h/week doing operative work (protection, inspection, and vigilance); these data were consistent with those reported by Sánchez-Nieto et al., Beltran et al., and Mexico’s National Institute of Statistics and Geography (INEGI) [36,39,40].

The nature of police work makes them deal with many stressors (the use of force, decision-making in critical situations, risks to one’s safety and that of colleagues, attending the scenes of accidents and injuries, and exposure to suffering and violence) that lead to a high risk of distress, affecting their health as well as their families in addition to the organization and the public [41,42]. Regarding the health of the police in Mexico, the self-perceived status is poor; however, there are no previous data on Mexico. Our study revealed that 43.17% of individuals perceive themselves to have a poor health status, 19.68% report having a chronic disease, 20.92% have digestive disorders, 17.27% have hypertension, and 9.24% have diabetes. Guimaraes et al. (2018) found increased cardiovascular and metabolic risk in police officers compared to office workers in Brazil [43]; in China, Yang et al. (2013) reported that health problems in Guangzhou police officers are serious, especially chronic, non-infectious diseases [44].

Among health status, nutrition is one of the most relevant factors. Regarding nutritional status, to our knowledge, this study was the first to evaluate the perception of the quality of diet, where 68.95% of police workforces consider their diet to be of poor quality; moreover, 84.36% do not have regular mealtimes. Zarrimpar et al. (2016) established that the disruption of cyclical expression of the circadian clock and key metabolic regulators through incorrect eating patterns contributes to obesity and dysmetabolism. Furthermore, the disruption of light/dark cycles leads to the same outcomes [45]. Chang et al. (2015) found that Taiwanese police officers that slept less than 5 h were 88% more likely to suffer abdominal obesity; this may explain the high prevalence rates of overweight and obesity found among the police workforce reported in various studies. The US National Health Interview Survey reported that workers in high-stress occupations, such as police officers and correctional security officers, had the second-highest prevalence of obesity among 41 fields, out of which law enforcement workers were obese.

Burnout is a tridimensional syndrome that affects people-related workers with high psychological requirements [5,46]; however, there are different criteria for an individual diagnosis of BO; Maslach and Jackson (1981) characterized the syndrome in individuals with three dimensions [5]. Considering scalar values of the MBI-HSS, our population presents medium levels of burnout in emotional exhaustion (23.29) and personal achievement (34.63), similar to data (35.07 +/−10.8) reported by Sánchez-Nieto (2011). Additionally, we found high scalar mean values for depersonalization (11.38) when compared to those noted for the Mexico City Police (4.55 +/−5.1) [36].

Regarding the prevalence of BO, we found a prevalence of 23.63% in Mexican police workforces. On the other hand, Guimarães et al. (2014) reported a prevalence of 56% in a population of police and military in Brazil [43], while Aranda and Pando (2010) reported 51.9% had more than one high dimension in Mexico [39], and Sánchez-Nieto (2011) reported a prevalence of BO of 44.6% [36]; this is because previous reports determined a positive BO case on the basis of at least one of the three dimensions being high.

Score values analyzed BO dimensionality. According to our data, 44.16% of the population presents high levels of EE, 49.29% present high levels of DP, and 41.03% present high levels of PA. Aranda and Pando (2010) found a prevalence of 12.3% in EE, 16.3% for DP, and 48.2% for PA [39], while Sánchez-Nieto (2011) reported 6.6% for EE, 16.8% for DP, and 36.9% for PA [36]. Although the data of the present investigation do not coincide with the dimensionality of BO in Mexico City police officers (PA > DP > EE), we agree with Hernández et al.’s (2006) theory about depersonalized behavior caused by emotional exhaustion in prison staff in Spain [47].

There are at least six models for BO reviewed by Montero-Martin and García-Campayo that attempt to explain the development of the syndrome [48]. One of the most accepted is the Leiter (1993) model, which proposes that in BO, the most critical aspect is the feelings of EE. As these resources are lost, they cannot give more of themselves at a certain level [49]. Depersonalization or negative attitudes toward clients appear due to the deterioration of the first dimension. At the same time, the lack of recognition or feedback at work leads to a lack of Personal Fulfillment.

Associations between BO and Sociodemographic, Health, and Nutritional Factors:

This study found weak direct correlations between EE and DP with seniority and BMI and a contrariwise correlation when using PA. Age failed to correlate to any dimension.

When we compared the mean values of each dimension for sociodemographic factors, we found lower values of DP in married people than in those who were single or divorced. The stability of marriage contributes to lowering the development of dehumanized and cynical attitudes toward people, feelings of being emotionally overextended by others, and contributes to a sense of achievement in work.

The computed model using MLP-ANN on Mexican Police officers found that two of the more critical factors related to BO were BMI and the self-perception of health status. Our data found that in Mexico, 8 of 10 workers of the police forces are overweight or obese; BMI correlates with all three dimensions of BO. Moreover, people at risk of the syndrome had the worst BMI (30.87 ± 6.81) compared to trait or normal BO (28.97 ± 3.94, *p* = 0.026 and 27.58 ± 3.17, *p* = 0.001). The relationship between Burnout Syndrome and BMI has been studied without a clear association or sufficient evidence [50]. Although Burnout Syndrome was initially described as untreated chronic stress related to work [5], despite physiological body responses to BO being unclear, organisms’ responses to stress involve cardiovascular, respiratory, endocrine, gastrointestinal, muscular, and reproductive systems. The sympathetic nervous system prepares the body for homeostasis disruption, increasing the heart rate; the respiratory system fails to supply oxygen to the body due to the shortness of breath caused by the constriction of the airways; meanwhile, the gastrointestinal system slows down digestion, and the absorption of nutrients is affected [51,52]. The hypothalamic–pituitary–adrenal (HPA) axis responds to stress by releasing glucocorticoid hormones. Cortisol increases the mobilization of glucose, free fatty acids, and amino acids by activating glycogenolysis and gluconeogenesis and suppressing insulin. Besides these effects, glucocorticoids increase appetite and food intake and may increase fat mass, leading to overweight or obesity during chronic stress.

Chronic stress dysregulates the HPA axis, causing cardiovascular diseases, hypertension, gastrointestinal disorders, and the downregulation of the immune responses. The self-perception of health status was also an important factor to BO with a direct relation; we found that people with the worst self-perception of health status presented high EE and DP levels and lowered PA levels. Meanwhile, people with the best self-perception of health had low EE levels and medium levels of DP and PA. The worst health status was related to the worst emotionally overextended feelings, as explained by Peterson et al. (2008), where burnout was associated with a poorer self-perception of health [53]. When we analyzed data of people with chronic diseases, hypertension, or diabetes, we found high levels of EE vs. medium levels in people without them; DP does not present any differences. This could be explained by the fact that chronic stress is enclosed in the EE dimension of BO; when EE increases, HPA axis regulation is affected. In 2013, Gómez et al. reported differences in BO in people with somatic and psychological comorbidity [54].

For the first time in Mexican police workforces, we studied how self-perception of variables that affect the circadian cycle (diet quality, regular mealtimes, and sleep hours) were related to BO. When individuals have poor diets, without regular mealtimes, and inadequate sleep hours, we find the worst levels of BO dimensions. The EE and DP were high in people with no control of the circadian cycle versus low or medium levels in controlled people. For PA, we found the worst levels with dysregulated diets and mealtimes, but not with the number of hours slept. Apparently, in police workforces, bad meal quality, mealtimes, and sleep hours habits lead to the progressive loss of work energy, frustration in relationships with others, and a loss of the sense of professional success. In addition, the disruption of circadian cycles, as demonstrated by No, J. (2018), can contribute to obesity in police officers [55,56].

Various authors point out that it is necessary to address the complexity of the work and psychosocial context and the characteristics of the individual and the work environment to explain BO. From this perspective, burnout could relate to specific factors associated with the type of occupation, which would mean different types of burnout or professional burnout, varying in etiology and symptoms [57,58,59,60].

However, the present research has some limitations. Sampling was non-probability instead of cluster sampling. The MBI-HSS was self-applied using a survey website instead of peer-to-peer, possibly leading to the misunderstanding of items; also, statistical methods could be improved by using network analysis.

This study found that in Mexican police workforces, Burnout Syndrome is dimensionally different than in all others work collectives previously studied.

## 5. Conclusions

Our findings support previous literature suggesting that policing is an occupation with multiple stressors and that these factors compromise the physical and mental health of police officers. However, to date, studies on the labor and health conditions of police officers in Latin America are scarce and inconsistent. Therefore, subsequent investigations must address the work environment and the organizational factors associated with the development of work stress in police officers, which could characterize a specific professional burnout based on the peculiarities derived from the performance of the police function.

We found a prevalence of Burnout Syndrome in 23.36% of the Mexican police workforce, which is associated with a poor self-perceived health status, age, diet quality, and a lack of regular mealtimes and sleep times. However, according to a multilayer analysis, BMI is the most critical factor related to the syndrome.

## Figures and Tables

**Figure 1 ijerph-19-05537-f001:**
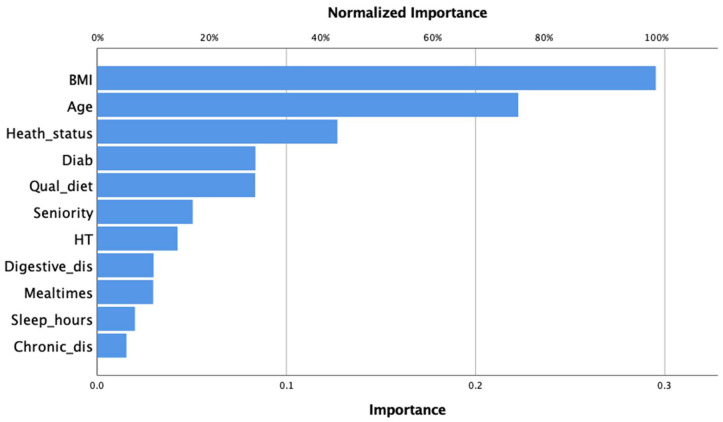
Importance of each factor related to BO computed by MLP-ANN in Mexican police officers.

**Table 1 ijerph-19-05537-t001:** MBI-HSS cut points.

Dimension	Items	Scores Cut Point		
		Normal	Trait	Risk
Emotional Exhaustion (EE)	1,2,3,6,8,13,14,16,20	0–18	19–26	27–54
Depersonanization (DP)	5,10,11,15,22	0–5	6–9	10–30
Personal achievement (PA)	4,7,9,12,17,18,19,21	0–33	34–39	40–48

**Table 2 ijerph-19-05537-t002:** Characteristics of the studied population.

Characteristic	Data
Age, mean (SD) years	39.93 (8.66)
Seniority, mean (SD) years	11.64 (8.4)
Body Mass Index, mean (SD)	29.12 (4.75)
Education	% (CI 95% lower–upper)
High School	16.13 (11.52–20.74)
College	60.48 (54.36–66.61)
Bachelor	20.56 (15.5–25.63)
Postgraduate	2.82 (0.75–4.9)
Marital status	
Single	14.52 (10.1–19.89)
Married	73.39 (67.85–78.93)
Divorced	12.1 (8.01–16.18)
Sleeping Hours	
Less than 4 h	16.47 (11.83–21.1)
4 to 6 h	66.67 (60.77–72.56)
6 to 9 h	16.87 (12.18–21.55)
Work Shifts	
8 h	10.04 (6.28–13.8)
12 × 12 h	10.44 (6.62–14.27)
12 × 24 h	14.86 (10.41–19.31)
24 × 24 h	55.82 (55.82–62.03)
Confined	4.01 (1.56–6.47)
others	4.82 (2.14–7.5)
Jurisdiction	
Municipal	54.84 (48.6–61.08)
State	36.29 (30.26–42.32)
federal	8.47 (4.98–11.96)
Other	0.4 (0–1.2)
Functions,	
Operative	85.89 (81.52–90.25)
Administrative	14.11 (9.75–18.48)
Health Status	
Poor Health Perc.	43.17 (35.99–48.34)
Chron. Diseases	19.68 (14.71–24.65)
Hypertension	17.27 (12.54–22.00)
Diabetes	9.24 (5.62–12.86)
Nutrition Status	
Poor Diet Quality	68.95 (61.77–75.13)
Regular Mealtimes	15.66 (11.12–20.21)
Digestive Disorders	30.92 (25.14–36.7)

**Table 3 ijerph-19-05537-t003:** Prevalence of Burnout Syndrome and dimensionality in police officers in Mexico.

	Scale	Score		
	Mean (SD)	Normal	Trait	Risk
		% (CI 95% Lower–Upper)	% (CI 95% Lower–Upper)	% (CI 95% Lower–Upper)
Burnout Syndrome	69.35 (22.17)	18.73 (14.65–22.85) ^a^	57.92 (52.77–63.14) ^b^	23.36 (18.91–27.81) ^c^
Dimensions				
Emotional Exhaustion	23.29 (0.89)	43.02 (37.82–48.22)	12.82 (9.31–16.34)	44.16 (38.94–49.38)
Depersonalization	11.38 (0.58)	34.76 (29.75–39.76)	15.95 (12.1–19.8)	49.29 (44.03–54.54)
Personal Achievement	34.62 (0.55)	38.18 (33.07–43.28)	20.8 (16.53–25.06)	41.03 (35.85–46.20)

Scale is the sum of values of items related to each dimension; all items were used for burnout. ^a^ “Normal” case of BO was considered if three dimensions were at low risk. ^b^ “Trait” case was considered if an individual had mixed scores. ^c^ “Risk” case was considered when three dimensions of BO were high.

**Table 4 ijerph-19-05537-t004:** Correlation of age, seniority, BMI, self-perception of health status, and self-perception of diet quality, with three dimensions of Burnout Syndrome in Mexican police officers.

	Emotional Exhaustion	Depersonalization	Personal Achievement
Age	−0.192	−0.179	0.189
Seniority	0.252	0.213	−0.226
BMI	0.219	0.152	−0.228
Health Status	−0.569	−0.555	0.387
Diet Quality	−0.441	−0.428	0.351
Since age and seniority were correlated, age was corrected for seniority, and seniority was corrected for age. All correlations were significant at the level of 0.01			

**Table 5 ijerph-19-05537-t005:** Mean difference analysis of health variables with three dimensions of Burnout Syndrome in Mexican police officers.

		Emotional Exhaustion			Depersonalization			Personal Achievement		
		Mean (SD)	Mean diff	*p*	Mean (SD)	Mean diff	*p*	Mean (SD)	Mean diff	*p*
Chronic diseases	Presence	31.20 (15.26)	8.464	0.002	16.22 (8.94)	5.204	0.001	32.29 (9.24)	−3.354	0.036
	Absence	22.74 (17.38)			11.02 (9.89)			35.64 (10.13)		
Hypertension	Presence	30.44 (17.25)	7.296	0.012	15.95 (10.28)	4.725	0.004	32.21 (10.75)	−3.349	0.046
	Absence	23.15 (17.07)			11.23 (9.66)			35.56 (9.8)		
Diabetes	Presence	31.78 (15.15)	8.128	0.031	14.65 (9.25)	2.873	0.186	31.87 (8.91)	−3.427	0.119
	Absence	23.65 (17.35)			11.78 (9.96)			35.30 (10.1)		
Digestive disorders	Presence	33.68 (15.88)	13.447	<0.001	17.43 (9.52)	7.768	<0.001	31.23 (10.33)	−5.468	<0.001
	Absence	20.23 (16.34)			9.66 (9.15)			36.70 (9.46)		
Regular mealtimes	Yes	18.67 (16.64)	−6.805	0.024	8.51 (8.86)	−4.187	0.015	39.13 (9.59)	4.919	0.005
	No	25.47 (17.23)			12.70 (9.98)			34.21 (9.94)		
Sleep quality	<4 h	30.63 (16.02)	5.345, 15.801 *	0.16	15.95 (9.5)	3.668, 8.666 *	0.076	32.37 (10.32)	−1.761, −8.539 *	0.551
	4–6 h	25.29 (16.88)	−5.345, 10.456 *	0.16	12.28 (9.76)	−3.668, 4.997 *	0.076	34.13 (9.95)	1.761, −6.778 *	0.551
	6–9 h	14.83 (16.37)	−15.801 *, −10.456 *	0.001	7.29 (9.16)	−8.666 *, −4.997 *	<0.001	40.90 (7.77)	8.539 *, 6.778 *	<0.001

Note: The tests assume equal variances. Sleep quality one-way ANOVA tests are fitted for all pairwise comparisons within a column of each innermost subtable using the Tukey correction. (*) indicates statistically significant differences.

## Data Availability

The data presented in this study are available on request from the corresponding author. The data are not publicly available due to ethical and privacy issues.

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
