# Peer review of "Burnout Syndrome and Related Factors in Mexican Police Workforces"

_ijerph, 2022, doi:10.3390/ijerph19095537_

Round 1

Reviewer 1 Report

This manuscript presents a study of burnout in Mexican police officers using the MBI instrument.

A description of the power analysis used to select the number (351) of officers included in the study should be included in the materials and methods.

Seniority and age would seem to be inherently codependent (as older people would likely have more seniority). How does correcting for age change the correlations with seniority (and vice versa)?

Minor comments

Line 15 – depersonalization is defined by the abbreviation “EP” here, but DP is used through the rest of the manuscript.

Line 66 – Start a new paragraph for the “Materials and Methods” section

Line 76 – the link https://es.surveymonkey.com/r/NK5VZBY is no longer functional. It may be more appropriate to include the survey instrument as a supplemental file in this manuscript rather than linking to SurveyMonkey.

Line 85/95 – AE is used in place of EE for emotional exhaustion

Line 97 appears to be a subheading. Please check to ensure the formatting conforms with the journal style.

Line 117-119 – It is unclear what is meant by “nine years of scholarly,” and “professional studies.” Please clarify.

Table 2 – Seniority is spelled without ñ in English.

Line 187 – Please rephrase “the average national scholarly”

The manuscript is readable but would be improved by thorough grammar editing.

Author Response

Dear reviewer,

We thank all your observations and commentaries made to our manuscript; all suggestions were essential to improving our work.

According to your revision, we have made some changes in our manuscript.

This manuscript presents a study of burnout in Mexican police officers using the MBI instrument.

A description of the power analysis used to select the number (351) of officers included in the study should be included in the materials and methods.

Response: Thanks for the observation. Now, on L152-161, we have described that we use the EpiInfo software Sample Size Calculator and the procedure to do it.

Seniority and age would seem to be inherently codependent (as older people would likely have more seniority). How does correcting for age change the correlations with seniority (and vice versa)?

  1. Thanks for this. As you said, seniority correlates to age (0.603, p<0.001). Now, we corrected table 4.

Minor comments

Line 15 – depersonalization is defined by the abbreviation “EP” here, but DP is used through the rest of the manuscript.

  1. Thanks, we appreciated the correction.

Line 66 – Start a new paragraph for the “Materials and Methods” section

  1. Thanks, we appreciated the correction.

Line 76 – the link https://es.surveymonkey.com/r/NK5VZBY is no longer functional. It may be more appropriate to include the survey instrument as a supplemental file in this manuscript rather than linking to SurveyMonkey.

  1. Thanks, we appreciated the suggestion. Now, the survey is included as a supplemental file.

Line 85/95 – AE is used in place of EE for emotional exhaustion.

  1. Thanks, we appreciated the correction.

Line 97 appears to be a subheading. Please check to ensure the formatting conforms with the journal style.

  1. Thanks, we appreciated the correction.

Line 117-119 – It is unclear what is meant by “nine years of scholarly,” and “professional studies.” Please clarify.

  1. Thanks, now we have reviewed the Education variable. Nine years of schooling means College studies, and Professional studies indicate Bachelor studies.

Table 2 – Seniority is spelled without ñ in English.

  1. Thanks, we appreciated the correction.

Line 187 – Please rephrase “the average national scholarly”.

  1. Thanks, we appreciated the correction.

The manuscript is readable but would be improved by thorough grammar editing.

  1. Thanks, we appreciated the observation; now manuscript has been reviewed by an English native speaker.

Reviewer 2 Report

The manuscript addressed an interesting topic for public health and occupational health. However, several concerns are raised on its justification, objectives, results and discussion.     

The background cited in the introduction is insufficient to introduce the topic, and the study lacks a strong justification. The authors mentioned general aspects related to "work stress" an burnout, but little is presented concerning BO in police workers. The introduction is short. It would be necessary to describe the context of police work, previous studies related to the theme (there are plenty) and to indicate the reasons to study this subject in this specific target population in Mexico. Also, the objectives are not described in any section. Therefore, this central issue is not clear along the whole text. 

The methods would also need a deep revision. Sampling procedures and the selection of participants should be detailed. A strongly biased sample may compromise study results. Any procedure was done to certify the representativeness of the sample? If not possible, what are the limitations of the findings? This should be further discussed. However, limitations are not discussed in the discussion section at all. More detailed information is also needed on the statistical methods the authors employed.

Regarding the results, considering the convenience non-representative sample, it might not be appropriate to estimate confidence intervals and to use statistical inference on collected data. Some tables are confusing and would need improvement. All results are descriptive, and confounding was not taken in consideration in any analyses.

The discussion is weak, and each finding should be more deeply explored, considering previous knowledge and an in-depth reflection on the reasons, context and possible perspectives regarding the most interesting results. The study has several limitations, and they should be discussed.

Lastly, a full revision of the manuscript, as well as possible additional analyses would need to be performed before considering the manuscript for publication. The text would also need an extensive English review.

Author Response

Dear reviewer,

We thank all your observations and commentaries made to our manuscript; all suggestions were essential to improving our work.

According to your revision, we have made some changes to our manuscript.

The manuscript addressed an interesting topic for public health and occupational health. However, several concerns are raised on its justification, objectives, results and discussion.     

The background cited in the introduction is insufficient to introduce the topic, and the study lacks a strong justification. The authors mentioned general aspects related to "work stress" an burnout, but little is presented concerning BO in police workers. The introduction is short. It would be necessary to describe the context of police work, previous studies related to the theme (there are plenty) and to indicate the reasons to study this subject in this specific target population in Mexico. Also, the objectives are not described in any section. Therefore, this central issue is not clear along the whole text. 

  1. Thanks for the observation; now, we have described the context of police work and previous studies and included L55 to L150.

The methods would also need a deep revision. Sampling procedures and the selection of participants should be detailed. A strongly biased sample may compromise study results. Any procedure was done to certify the representativeness of the sample? If not possible, what are the limitations of the findings? This should be further discussed. However, limitations are not discussed in the discussion section at all. More detailed information is also needed on the statistical methods the authors employed.

  1. Thanks for the observation; now, on L152-161, we have described that we use the EpiInfo software Sample Size Calculator and the procedure to do it. In addition, limitations have been added in L368-371.

Regarding the results, considering the convenience non-representative sample, it might not be appropriate to estimate confidence intervals and to use statistical inference on collected data. Some tables are confusing and would need improvement. All results are descriptive, and confounding was not taken in consideration in any analyses.

  1. Thanks for the observation, table 2 was simplified, tables were simplified, and partial correlations were calculated for age-seniority and BO dimensions.

The discussion is weak, and each finding should be more deeply explored, considering previous knowledge and an in-depth reflection on the reasons, context and possible perspectives regarding the most interesting results. The study has several limitations, and they should be discussed.

  1. Thanks for the observation; we reviewed the discussion; we hope now this research contributes more to this matter.

Lastly, a full revision of the manuscript, as well as possible additional analyses, would need to be performed before considering the manuscript for publication. The text would also need an extensive English review.

  1. Thanks for the suggestion, now, a Network Analysis of BO and related factors are presented, and English has been reviewed.

Reviewer 3 Report

The objective of this study is to evaluate the association between Burnout Syndrome and sociodemographic, occupational, and health factors in Mexican police officers. The study founds a prevalence of 65.06% burnout syndrome in the Mexican police work 289 forces, meanly associated to a poor self-perceived health status, diet quality, lack of regu-290 lar mealtimes, and sleep times.

The methods are well presented

I would suggest to report the results (lines 164/175) in a table form.

Table 2: Señority, to be changed in seniority

I suggest an English spelling and grammar check. 

Few typing errors

Author Response

Dear reviewer,

We thank all your observations and commentaries made to our manuscript; all suggestions were essential to improving our work.

According to your revision, we have made some changes to our manuscript.

I would suggest to report the results (lines 164/175) in a table form.

  1. Thanks, now we added a new table with these data.

Table 2: Señority, to be changed in seniority

  1. Thanks, we appreciated the correction.

I suggest an English spelling and grammar check. 

  1. Thanks, we appreciated the observation; now manuscript has been reviewed by an English native speaker.

Few typing errors

  1. Thanks, we appreciated the observation; now, we have reviewed the manuscript looking for typing errors.

Reviewer 4 Report

Dear Authors,

Thank you for the opportunity to read your relevant work about burnout in the police workforce.

Burnout syndrome is a huge issue for the police workforce. However, it is pretty challenging to clearly know the antecedents and consequents of the relationship between burnout and health status, as low health status increases burnout and vice versa.

What is the hypothesis of the study?

What are the gaps the authors want/will cover? The statement that there are few studies in the area is not enough to justify the study.

There are some format issues, line 66 materials and methods in the middle of the text.

Although the data is interesting, the study is descriptive.

The data described between lines 162 up to discussion could be shown in graphs.

There are issues with the citations.

Dear authors, I would like the comments I will make to be interpreted in the sense of helping to think about the work and better ways to make use of the data that is interesting.

The first comment is that burnout is only possible to see its prevalence when the three dimensions in the same individual are high; that is, he/she has to combine high emotional exhaustion, and high depersonalization, and low personal achievement.

The data described do not separate the groups into normal, trait and risk in burnout syndrome because the data should be treated together, therefore, not each dimension separately. By treating them separately, what we have are isolated variables, but we do not reach the syndrome as it was proposed by Maslach. Therefore, I suggest that the authors revise their proposed analyses to achieve coherence with the work´s aim: “The objective of this study was to evaluate the association between Burnout Syndrome and sociodemographic, occupational, and health factors in Mexican police officers.”

Considering that one of the variables the authors have in the study is the number of hours of recovery, the study could draw on this variable to study it as a moderator of the relationship between recovery and health variables. In other words, advance the model by proposing a study on variables of protection (recovery) to the policeman's health. Other studies in Brazil have already pointed out the protective role of recovery in the police workforce and could help.

Another alternative to reinterpret the data has been used in health since health data, besides being multicausal, are mutually related (i.e. it is not known how much psychological variables predict physiological variables or vice versa), and one could present a network analysis where one can see mediators between the different variables from what the data shows.

Again, it is necessary to group the burnout variable to understand the prevalence of the syndrome in the population. To advance in this sense, it would be possible to perform a latent class profile analysis.

The point is that a descriptive paper, as it is in the present study, contributes very little to what one might expect for such a study.

I hope I have been helpful with the comments.

Author Response

Dear reviewer,

We thank all your observations and commentaries made to our manuscript; all suggestions were essential to improving our work.

According to your revision, we have made some changes in our manuscript.

Thank you for the opportunity to read your relevant work about burnout in the police workforce.

  1. Thank you for all the valuable commentaries on our work; we appreciate all the suggestions.

Burnout syndrome is a huge issue for the police workforce. However, it is pretty challenging to clearly know the antecedents and consequents of the relationship between burnout and health status, as low health status increases burnout and vice versa.

What is the hypothesis of the study?

  1. Thanks for the observation; we hypothesize that BO in Mexican police forces is higher than in other countries, and health variables correlate to the worst values on BO dimensions.

What are the gaps the authors want/will cover? The statement that few studies in the area is not enough to justify the study.

  1. Authors want to know the sociodemographic, occupational, and health factors related to BO dimensions in Mexican Police workforces.

There are some format issues, line 66 materials, and methods in the middle of the text.

  1. Thanks for the observation; this issue was corrected.

Although the data is interesting, the study is descriptive.

  1. Thanks for the observation; now, we added to table 5 were compared the scale values of each BO dimension in each health variable.

The data described between lines 162 up to discussion could be shown in graphs.

  1. Thanks for the observation; now we added to table 5 as you suggested.

There are issues with the citations.

  1. Thanks for the observation; we have reviewed all the citations.

Dear authors, I would like the comments I will make to be interpreted in the sense of helping to think about the work and better ways to make use of the data that is interesting.

  1. Thanks for your invaluable commentaries and uninterested help to our work.

The first comment is that burnout is only possible to see its prevalence when the three dimensions in the same individual are high; that is, they have to combine high emotional exhaustion, and high depersonalization, and low personal achievement.

  1. Thanks for the suggestion; now, we consider it positive for BO when all three dimensions have a high score.

The data described do not separate the groups into normal, trait, and risk in burnout syndrome because the data should be treated together, therefore, not each dimension separately. By treating them separately, what we have are isolated variables, but we do not reach the syndrome as it was proposed by Maslach. Therefore, I suggest that the authors revise their proposed analyses to achieve coherence with the work´s aim: “The objective of this study was to evaluate the association between Burnout Syndrome and sociodemographic, occupational, and health factors in Mexican police officers.”

  1. Thanks for the suggestion; now, we consider positive for BO when all three dimensions have a high score.

Considering that one of the variables the authors have in the study is the number of hours of recovery, the study could draw on this variable to study it as a moderator of the relationship between recovery and health variables. In other words, advance the model by proposing a study on variables of protection (recovery) to the policeman's health. Further studies in Brazil have already pointed out the protective role of recovery in the police workforce and could help.

  1. Thanks for the invaluable suggestion, authors declined this, and we tried the Network Analysis.

Another alternative to reinterpret the data has been used in health since health data, besides being multicausal, are mutually related (i.e. it is not known how much psychological variables predict physiological variables or vice versa), and one could present a network analysis where one can see mediators between the different variables from what the data shows.

  1. Thanks for the invaluable suggestion; now, a Network Analysis of BO and related factors is presented.

Again, it is necessary to group the burnout variable to understand the prevalence of the syndrome in the population. To advance in this sense, it would be possible to perform a latent class profile analysis.

  1. Thanks for the suggestion; now, we consider positive for BO when all three dimensions have a high score.

The point is that a descriptive paper, as it is in the present study, contributes very little to what one might expect for such a study.

  1. Thanks for the observation; the authors hope that we could contribute more to this important matter with a new table and new analysis.

Round 2

Reviewer 4 Report

Dear authors,

Thank you for the new version of your work. From my point of view, it improves in many directions.

The work is relevant; the police force is less studied compared to teachers and health-system concerning burnout syndrome.

There are some issues in the citation and references in the newly added sections; I suggest reviewing them.

The aim of the work is clear; however, you could posit between lines 101 and 102 the contribution of your work.

I suggest adding in Table 2 the whole name of BMI (Body Mass Index); it is the only factor with no complete expression.

I suggest adding in Table 3 the general percentage. Burnout is the combination of the three dimensions and not each other separately. To describe the prevalence of burnout is the combination of high-high-high risk percentage, and normal is the low-low-low. It is possible to assume that in line 197 what you describe (23.36%) is the high-high-high percentage, but it is not clear.  

I suggest in Table 4 adding the “Health Status self-perception” and “Quality of Diet self-perception” as they use a 5-point scale; it is easy to understand applying an average and SD.

The adding information on multilayer perception (very interesting analysis, I congrats the authors), in the description done (lines 243-245), I suggest being clear about BMI, age, health status etc. For example, a higher BMI is the most important and a less age, and low health status and… As the direction of the variable change, it is clear to state in this way.

In the conclusion, line 363, it is important to say that BMI is also a problem as it is the most critical factor in the analysis.

Thank you very much for the opportunity to read your work.

Author Response

Dear reviewer, 

We thank all your observations and commentaries made to our manuscript; all suggestions were essential to improving our work.

According to your revision, we have made some changes in our manuscript.

Dear authors,

Thank you for the new version of your work. From my point of view, it improves in many directions.

  1. Thank you for your kindly suggestions; it was constructive to improve our work.

The work is relevant; the police force is less studied compared to teachers and healthsystem concerning burnout syndrome.

  1. Yes, one matter not discussed in this work is that an important police association in  Mexico is involved. They are learning how to study this syndrome, and they are now open to collaborations.

There are some issues in the citation and references in the newly added sections; I suggest reviewing them.

  1. Thanks for the suggestion; we reviewed the manuscript citations and references.

The aim of the work is clear; however, you could posit between lines 101 and 102 the contribution of your work.

  1. Thanks, now he has added a couple of lines with the contribution.

I suggest adding in Table 2 the whole name of BMI (Body Mass Index); it is the only factor with no complete expression.

  1. Thanks, now we changed BMI to Body Mass Index in Table 2.

I suggest adding in Table 3 the general percentage. Burnout is the combination of the three dimensions and not each other separately. To describe the prevalence of burnout is the combination of high-high-high risk percentage, and normal is the low-low-low. It is possible to assume that in line 197 what you describe (23.36%) is the high-high-high percentage, but it is not clear.  

  1. Thanks for the kind suggestion. We added the BO results to table 3 and annotated that a BO case is considered at risk when all three dimensions are at risk. We understand the importance of treating BO as a combination of dimensions; despite this, some research on police forces in south America considers a positive case of BO if one of the dimensions is high. In order of comparison, we report each dimension.

I suggest in Table 4 adding the “Health Status self-perception” and “Quality of Diet selfperception” as they use a 5-point scale; it is easy to understand applying an average and SD.

  1. Thanks for the kind suggestion. We added the Health Status and Diet Quality selfperception to table 4.

The adding information on multilayer perception (very interesting analysis, I congrats the authors), in the description done (lines 243-245), I suggest being clear about BMI, age, health status, etc. For example, a higher BMI is the most important and a less age, and low health status and… As the direction of the variable change, it is clear to state in this way.         R. Thanks for the suggestion. Now we have added this information to the MLP-ANN description.

In the conclusion, line 363, it is important to say that BMI is also a problem as it is the most critical factor in the analysis.

  1. Thanks for the suggestion. Now we have added this conclusion.

Thank you very much for the opportunity to read your work.